# Effects of Management Practices on Quinoa Growth, Seed Yield, and Quality

**Ning Wang [1], Fengxin Wang [1],*, Clinton C. Shock [2], Chaobiao Meng [1] and Lifang Qiao [1]**

[1] Center for Agricultural Water Research in China, China Agricultural University, Beijing 100083, China; ningwang@cau.edu.cn (N.W.); cau_mcb@cau.edu.cn (C.M.); qiaolf1995@163.com (L.Q.)

[2] Malheur Experiment Station, Oregon State University, 595 Onion Ave., Ontario, OR 97331, USA; clinton.shock@oregonstate.edu

* Correspondence: fxinwang@cau.edu.cn; Tel.: +86-1364-123-2337

**Abstract:** Quinoa (*Chenopodium quinoa* Willd.) yield potential needs to be further achieved by good management practices to meet the increasing global demand. Two years of orthogonal field experiments were undertaken to investigate the effects of irrigation onset criteria using soil matric potential (SMP) (−15, −25, and −55 kPa), nitrogen fertilizer rate (80, 160, and 240 kg ha$^{-1}$), and plant density (20, 30, and 40 plants m$^{-2}$) on quinoa growth, seed yield, weight, and protein content. Initiating irrigations at an SMP of −15 to −25 kPa achieved significantly ($p < 0.05$) greater seed yield (37.2 g plant$^{-1}$), thousand kernel weight (2.25 g), and protein content (21.2%) than −55 kPa (25.2 g plant$^{-1}$, 2.08 g, and 19.8%, respectively). The 240 kg ha$^{-1}$ nitrogen rate had significantly ($p < 0.05$) greater thousand kernel weight (2.26 g) and protein content (21.3%) than 80 (2.07 g and 19.5%, respectively) and 160 kg ha$^{-1}$ (2.14 g and 20.7%, respectively). The yield under 20 plants m$^{-2}$ reached 39.5 g plant$^{-1}$, which was 13.5 g plant$^{-1}$ higher than 40 plants m$^{-2}$ ($p < 0.05$). The quinoa consumed most of the available nitrogen in the soil (410–860 kg ha$^{-1}$), indicating that quinoa should be part of a sound crop rotation program.

**Keywords:** soil matric potential; nitrogen application rate; plant density; drip irrigation; nitrogen uptake

## 1. Introduction

Quinoa (*Chenopodium quinoa* Willd.) is a potential nutritious crop for human consumption [1–3]. Quinoa seed yields under moderate management were quite low, around 500–700 kg ha$^{-1}$ [4,5]. However, quinoa yield could be much higher under the proper management of irrigation (310–1300 mm) [6,7], nitrogen fertilization (120–180 kg ha$^{-1}$) [8,9], and plant density (10–300 plants m$^{-2}$) [5,10,11]. Therefore, higher quinoa yields may be attained by simultaneously optimizing irrigation, nitrogen fertilization, and plant density.

Although quinoa is a drought-tolerant crop with a low water requirement [12], quinoa seed yields respond positively to irrigation [13–15]. As an efficient water-saving irrigation method, drip irrigation has been widely applied in water-limited regions [16–18] and could be scientifically scheduled to improve quinoa production. Geerts et al. [14], Hirich et al. [19], Fghire et al. [20], and Rachid et al. [21] scheduled quinoa drip irrigation based on estimated crop evapotranspiration ($ET_C$) while completely ignoring the influence of the actual soil water status. Razzaghi et al. [22,23] grew quinoa relying on the measurement of soil water content, paying little attention to the dramatic spatial–temporal variations in soil water.

Soil matric potential (SMP) is a useful criterion for characterizing crop soil water availability, and SMP-based drip irrigation management has been successfully applied to improve yields in many

crops [24–29]. The SMP threshold for quinoa drip irrigation scheduling can vary with soil texture, active rooting depth of quinoa, planting configuration, water availability for irrigation, and many other factors. However, there is no reference in the literature to SMP irrigation onset for quinoa.

Nitrogen fertilizer should be important for quinoa because quinoa is high in protein content [9]. Quinoa seed yields generally increase with an increasing nitrogen rate [8,9,30]. The reported optimal nitrogen application rate varies widely by authors and locations: 120 kg ha$^{-1}$ in Germany [8], 180 kg ha$^{-1}$ in Denmark [9], and 310 kg ha$^{-1}$ in Egypt [30]. Notably, a small yield decrease was also observed when increasing the nitrogen application up to the highest nitrogen application of 160 kg ha$^{-1}$ [11]. These disparities can be understood by the large variations in soil fertility, varieties, and crop needs, as affected by water, nutrition supply, plant density, and other environmental constraints [31,32]. Little reported information on the nitrogen application management of quinoa considers nitrogen uptake and soil fertility.

Plant density is an important factor to ensure high quinoa seed yield [11,33–35], which in turn is influenced by many factors, like crop varieties, climate conditions, and cropping strategies [36,37]. Spehar and Rocha [35] showed that the plant density varying from 10 to 60 plants m$^{-2}$ did not influence seed yield when irrigated using sprinkling irrigation at approximately seven-day intervals in Brazil. Gimplinger et al. [34] found a quadratic response of seed yield to plant density, and the plant density of 17 plants m$^{-2}$ reached the maximum yield ($p < 0.05$) with no water and fertilizer supply in eastern Austria.

The objectives of this study were as follows: (1) to examine the effects of the SMP irrigation criteria, nitrogen application rate, and plant density on quinoa growth, seed yield and quality, and nitrogen uptake; and (2) to provide a scientific basis for irrigation, nitrogen fertilization, and plant density management for drip-irrigated quinoa production.

## 2. Materials and Methods

### 2.1. Experimental Site

The experiments were performed at the Shiyanghe Experimental Station of China Agricultural University in Wuwei, Gansu Province, China (37°52′ N, 102°50′ E, 1581 m altitude) during the growing seasons of 2018 and 2019. The experiments were carried out in a typical continental temperate climate with an annual average temperature of 8.8 °C, mean annual precipitation of 164 mm, mean total sunshine of 3000 h, and a frost-free season of 150 days. The experiments' field soil parameters are listed in Table 1.

**Table 1.** Soil parameters of 0–90 cm soil layers before fertilization at Wuwei, Gansu Province, China, 2018 and 2019.

| Soil Parameters | 2018 | 2019 |
|---|---|---|
| Soil bulk density (g cm$^{-3}$) | 1.5 | 1.5 |
| Clay (%) | 9.8 | 9.9 |
| Silt (%) | 64.6 | 65.2 |
| Sand (%) | 25.6 | 24.9 |
| Field capacity (%) | 30.9 | 30 |
| Mineral content (%) | 1.245 | 1.091 |
| Available nitrogen (mg kg$^{-1}$) | 51.6 | 60.8 |
| Available phosphorus (mg kg$^{-1}$) | 20.1 | 12.4 |
| Available potassium (mg kg$^{-1}$) | 261 | 128 |
| Soil electric conductivity (μs cm$^{-1}$) | 157.5 | 161.8 |

### 2.2. Experimental Design and Treatments

Referring to a review paper by Shock and Wang (2011) [29], the −25 kPa SMP was considered an optimum irrigation onset setting for many field crops, such as potato and corn; therefore, −25 kPa was

chosen as the intermediate irrigation level. To explore the performance of quinoa under non-stress and water stress conditions, −15 and −55 kPa were applied in our experiments, respectively. A nitrogen application rate of 80, 160, and 240 was adopted in these experiments, comparable to some other experiments [9,11,30]. As for the plant density, Yang (2015) [38] and Fen (2019) [39] recommended that the quinoa (c.v. Longli No.1, adopted in this study) should be about 10–20 plants $m^{-2}$ in Northwest China. We aimed to explore the possible greater yield under a higher plant density with adequate water and nitrogen supply; therefore, plant densities of 20, 30, and 40 plants $m^{-2}$ were applied in these experiments.

Since orthogonal design only considers a fraction of the combination of variables to investigate a wide range of operating conditions, it becomes one of the most time-saving experimental design methods for multiple-factor experiments [40]. Therefore, the three-factor and three-level orthogonal design was used in the field experiments both in 2018 and 2019 (Table 2). The treatments were replicated three times with a completely randomized design each year. The plot sizes were 24 $m^2$ (6 m (length) × 4 m (width)). There was a ridge (width: 20 cm) between adjacent plots.

**Table 2.** Experimental treatments for quinoa production in 2018 and 2019.

| Treatments | Soil Matric Potential (−kPa) | Nitrogen Rate (kg ha$^{-1}$) | Plant Density (plants m$^{-2}$) |
|---|---|---|---|
| T1 | 15 | 80 | 20 |
| T2 | 15 | 160 | 30 |
| T3 | 15 | 240 | 40 |
| T4 | 25 | 80 | 30 |
| T5 | 25 | 160 | 40 |
| T6 | 25 | 240 | 20 |
| T7 | 55 | 80 | 40 |
| T8 | 55 | 160 | 20 |
| T9 | 55 | 240 | 30 |

### 2.3. Agronomic Practices

All the phosphorus fertilizer (P, totaling 30 kg ha$^{-1}$), in the form of diammonium phosphate, was broadcast with sowing. Potassium fertilizer (K, totaling 180 kg ha$^{-1}$) in the form of potassium sulfate and nitrogen fertilizer (N) in the form of urea were split into two applications, half were broadcast with sowing and the remaining half was applied through the drip irrigation system at the beginning of the flowering stage (early July) in both years.

Thin-wall (0.2 mm) drip tapes (Beijing Lvyuan Plastic Co., Ltd., Beijing, China) with a flow rate of 2.5 L h$^{-1}$ per emitter were placed between each pair of rows. The emitter spacing was 20 cm. Each plot had an individual valve and a flow meter to measure independently the irrigation volume. Transparent plastic films 0.008 mm thick and 1.2 m wide were laid after the drip irrigation system was installed.

A locally recommended quinoa variety "Longli No.1" was planted on April 28 in both 2018 and 2019. This variety originated from Bolivian material (cv. Puno) and was adapted in China for middle maturity (100–130 days of growing season), high yield, and strong resistance to disease, cold, and drought [38,39].

Quinoa seeds were sown in pits (1–2 cm deep) by hand with planting spacing of 10 cm. According to the designed plant density, the row spacing was 50, 33, and 25 cm under 20, 30, and 40 plants $m^{-2}$ plant density treatments, respectively. Each plot of plant density of 20, 30, and 40 plants $m^{-2}$ treatments contained 12, 18, and 24 rows, respectively. The distances between the edge of the ridge and the outermost plant on the corresponding side of each plot were 25, 17, and 13 cm (half of the row spacing) under 20, 30, and 40 plants $m^{-2}$ plant density treatments, respectively. After germination, thinning was carried out two times to retain one plant per pit.

### 2.4. Irrigation Scheduling

All treatments received 50 mm irrigation water after sowing to assure uniform and rapid seed germination in both years. Then, irrigation water (27 mm) was applied when the SMP reached the corresponding onset criteria. The SMP was measured using tensiometers (WST–2B, Beijing Waterstar Tech. Co., Ltd., Beijing, China). All tensiometers were installed at 0.2 m depth. The soil water retention characteristics based on field measurements are presented in Figure 1, and the soil matric potential of −15, −25 and −55 kPa represented the volumetric soil water content of 23% (77% of field capacity), 18% (60% of field capacity) and 13% (43% of field capacity).

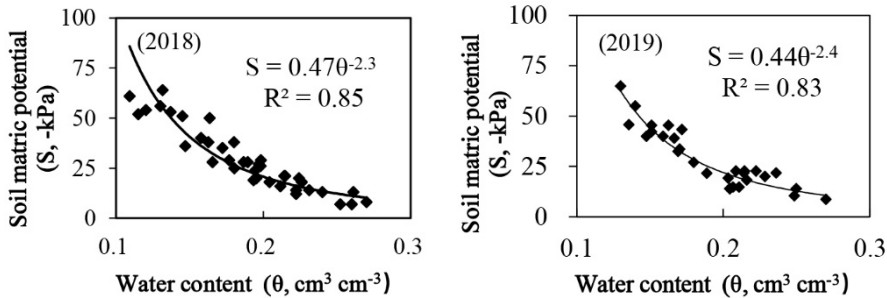

**Figure 1.** The soil water retention characteristics of experimental fields in 2018 and 2019.

### 2.5. Measurements and Calculations

#### 2.5.1. Estimation of Crop Evapotranspiration

Crop evapotranspiration ($ET_C$) was calculated using the soil water balance method [41]:

$$ET_C = I - K - \Delta W - S + R \tag{1}$$

where $I$ is the irrigation amount (mm), $K$ is deep drainage below crop root zone (mm), $\Delta W$ is the change of soil water storage (mm), $S$ is the surface runoff (mm), and $R$ is rainfall (mm).

$\Delta W$ can be calculated as follows:

$$\Delta W = S_{t1} - S_{t2} \tag{2}$$

where $S_{t1}$ and $S_{t2}$ are the water storage (mm) in the root zone before planting ($t_1$) and after harvest ($t_2$), respectively. To estimate $\Delta W$, soil water content in the soil profile was measured gravimetrically. The sampled depths were 0.1, 0.2, 0.3, 0.5, 0.7, and 0.9 m, respectively.

Precipitation events during the growing season were relatively small, and the designed depth of wetted soil layer ($H$) was far less than the maximum depth of soil water content measurement (0.9 m), therefore $K$ and $S$ should be negligible. The water contribution from groundwater was negligible as the groundwater table was below more than 25 m. Therefore, Equation (1) was simplified as follows:

$$ET_C = I - \Delta W + R \tag{3}$$

#### 2.5.2. Plant Height

Plant heights were measured on ten plants from the center of each plot from the ground surface to the tip of the inflorescence on the main stem at 49, 52, 59, 66, and 80 days after sowing (DAS) in 2018 and 51, 61, 68, 74, and 84 DAS in 2019.

#### 2.5.3. Leaf Area Index (LAI)

The leaf area index (LAI) was measured from the center of each plot using Sunscan (Beijing Aozuo Ltd., Beijing, China) at 55, 61, 76, and 89 DAS in 2018 and 52, 61, 68, 74, and 84 DAS in 2019.

### 2.5.4. Dry Matter and Protein Content of Leaf, Stem, and Seed, and Nitrogen Uptake

At physiological maturity in 2018 and 2019, six plants were sampled above ground from the center of each plot and were separated into stem, leaf, and seed. The samples were oven-dried at 85 °C until the mass did not change anymore. Nitrogen concentrations were determined by the Kjeldahl method [42]. Nitrogen uptake was the sum of the multiplication of dry matter and nitrogen content of stem, leaf, and seed (kg ha$^{-1}$).

### 2.5.5. Yield Per Plant

At the physiological maturity on August 18 (112 DAS) of both years, fifteen plants from the center of each plot were manually harvested. Seed samples were air-dried, threshed, and screened, and yield per plant (g plant$^{-1}$) was obtained and expressed at the 10%–12% moisture level (measuring by the oven-drying method).

### 2.5.6. The Thousand Kernel Weight and Seed Protein Content

The thousand kernel weight was measured counting out a thousand seeds three times per plot and weighing them on an electronic scale to the nearest 0.01 g.

Generally, the nitrogen (N) content of protein is around 16%; therefore, the seed protein content was estimated from the seed nitrogen content multiplied by 6.25 [9,42].

### 2.5.7. Soil Available Nitrogen Amount

The available nitrogen amount within a depth of 1 m before sowing can be calculated by the following equation:

$$AN = A \times \rho \times D \times \theta \tag{4}$$

where $AN$ is available nitrogen, $A$ is the area, $\rho$ is soil bulk density, $D$ is soil depth (1 m), and $\theta$ is available nitrogen content.

### 2.6. Statistical Analysis

The ANOVA was used in the variance analysis (*F*-test) to analyze the main effects of SMP, nitrogen application rate, and plant density on plant height, LAI, dry matter, seed yield, thousand kernel weight, protein content, and nitrogen uptake using SPSS 19.0 version (SPSS Inc., Chicago, IL, USA) [43]. The fixed factors were the irrigation, nitrogen rate, and plant density, and there was no random factor. The post hoc multiple comparison test and the equal variances were calculated by LSD (least significant difference) and S-N-K (Student-Newman-Keuls). In this study, the interactions among SMP, nitrogen application rate, and plant density were excluded and the statistical differences were significant when the *p*-value was less than 0.05.

## 3. Results

### 3.1. Weather Conditions and Insect Pressure

The daily rainfall and mean air temperature of the quinoa growth period during 2018 and 2019 are presented in Figure 2. Rainfall occurred 19 and 29 times totaling 107 and 125 mm in the 2018 and 2019 quinoa growing seasons, respectively. The mean daily air temperature fluctuation trends were similar both seasons, ranging from 10.6 to 27.3 °C and 8.5 to 25.0 °C during the 2018 and 2019 quinoa growing seasons, respectively.

Very little insect pressure occurred and insect control measures were not necessary.

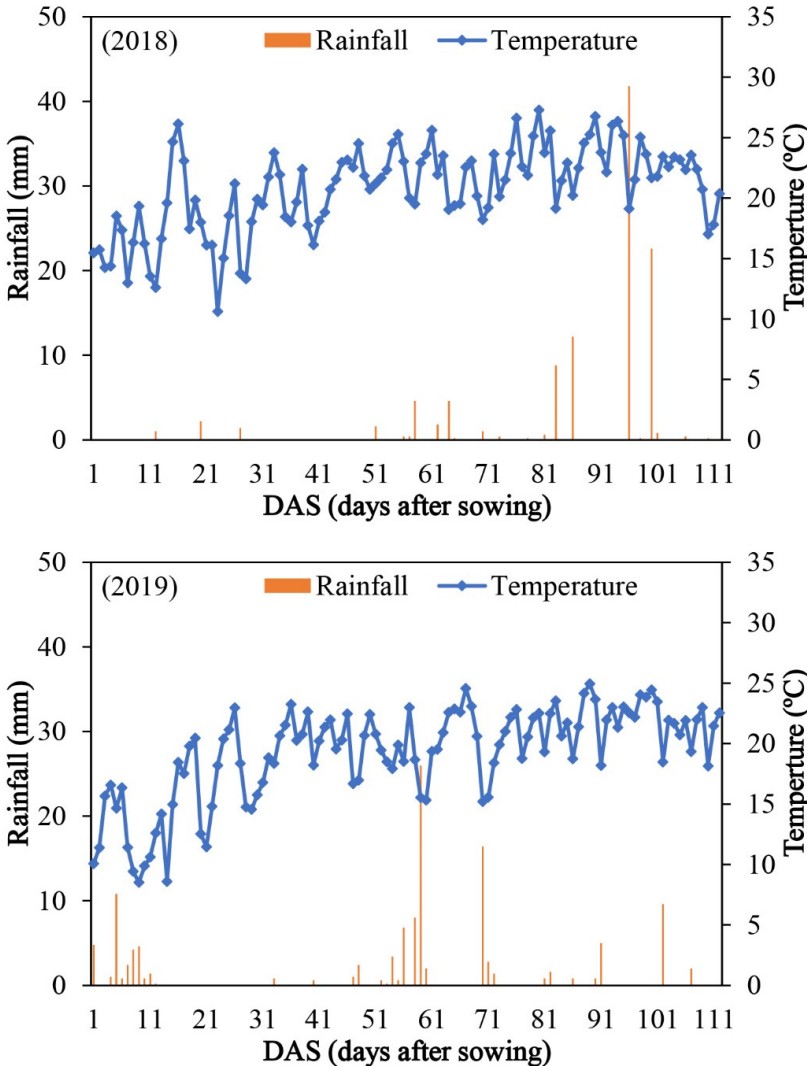

**Figure 2.** Rainfall and mean air temperature at 2.0 m height during the quinoa growing seasons in 2018 and 2019 at Wuwei, Gansu Province, China.

### 3.2. Quinoa Seasonal Evapotranspiration

Quinoa received significantly ($p < 0.05$) greater seasonal $ET_C$ under the wetter (less negative) SMP treatment in both years (Table 3). In 2018, the differences in $ET_C$ were significant ($p < 0.05$) between all three SMP treatments. In 2019, −55 kPa SMP had significantly ($p < 0.05$) lower $ET_C$ than those under −15 and −25 kPa SMP, whereas the difference between −15 and −25 kPa SMP was not significant ($p > 0.05$). Increasing the nitrogen application rate from 80, 160, to 240 kg ha$^{-1}$ increased the seasonal $ET_C$ in both years, but a significant ($p < 0.05$) difference was only found in 2018 (Table 3). Quinoa $ET_C$ became higher as plant density increased (Table 3). The seasonal $ET_C$ under 40 plants m$^{-2}$ was 479 mm and 440 mm, respectively in 2018 and 2019, significantly greater than that of 20 plants m$^{-2}$ treatment (382 mm and 365 mm, respectively in 2018 and 2019). The significant difference in $ET_C$ between plant density of 30 and 40 plants m$^{-2}$ was observed ($p < 0.05$) in 2018 but not in 2019.

### 3.3. Quinoa Height and Leaf Area Index (LAI)

Quinoa heights tended to increase as the SMP increased both in 2018 and 2019 (Table 4). The differences in quinoa height between −15 and −55 kPa SMP treatments were significant ($p < 0.05$) throughout the two growing seasons. Moreover, the differences in plant height between −15 and −25 kPa SMP were significant ($p < 0.05$) except for 59 DAS in 2018 and 51, 68, 74, and 84 DAS in 2019.

The nitrogen rate of 240 kg ha$^{-1}$ resulted in a significantly taller ($p < 0.05$) quinoa plant height than 80 kg ha$^{-1}$ treatments (Table 4). Quinoa height was not significantly affected by plant density in either growing season (Table 4).

**Table 3.** Effects of soil matric potential, nitrogen application rate, and plant density on quinoa crop evapotranspiration ($ET_C$), and yield during the 2018 and 2019 growing seasons.

| Treatment | | 2018 | | 2019 | |
|---|---|---|---|---|---|
| | | $ET_C$ | Yield | $ET_C$ | Yield |
| | | (mm) | (g plant$^{-1}$) | (mm) | (g plant$^{-1}$) |
| Soil matric potential (−kPa) | 15 | 534 [a] | 38.1 [a] | 495 [a] | 36.2 [a] |
| | 25 | 433 [b] | 31.5 [ab] | 424 [a] | 32.6 [a] |
| | 55 | 347 [c] | 25.6 [b] | 311 [b] | 24.8 [b] |
| Nitrogen rate (kg ha$^{-1}$) | 80 | 429 [b] | 29.8 [a] | 398 [a] | 28.4 [b] |
| | 160 | 430 [b] | 32.6 [a] | 413 [a] | 32.8 [a] |
| | 240 | 455 [a] | 32.4 [a] | 420 [a] | 32.3 [a] |
| Plant density (plants m$^{-2}$) | 20 | 382 [c] | 39.7 [a] | 365 [b] | 39.2 [a] |
| | 30 | 453 [b] | 33.3 [ab] | 426 [ab] | 33.1 [b] |
| | 40 | 479 [a] | 26.2 [b] | 440 [a] | 25.8 [c] |

Values followed by different small letters (a, b and c) indicate significant differences at $p < 0.05$.

**Table 4.** Effects of soil matric potential, nitrogen application rate, and plant density on quinoa height (cm) at 49, 52, 59, 66, and 80 days after sowing (DAS) in 2018 and at 51, 61, 68, 74, and 84 DAS in 2019.

| Treatment | | 2018 | | | | | 2019 | | | | |
|---|---|---|---|---|---|---|---|---|---|---|---|
| | | 49 | 52 | 59 | 66 | 80 | 51 | 61 | 68 | 74 | 84 |
| Soil matric potential (−kPa) | 15 | 96 [a] | 119 [a] | 151 [a] | 177 [a] | 179 [a] | 71 [a] | 121 [a] | 158 [a] | 178 [a] | 184 [a] |
| | 25 | 93 [b] | 115 [b] | 146 [ab] | 169 [b] | 173 [b] | 67 [ab] | 114 [b] | 151 [ab] | 169 [ab] | 171 [ab] |
| | 55 | 91 [c] | 112 [b] | 139 [b] | 166 [b] | 172 [b] | 62 [b] | 110 [b] | 141 [b] | 161 [b] | 165 [b] |
| Nitrogen rate (kg ha$^{-1}$) | 80 | 85 [c] | 106 [b] | 135 [b] | 158 [b] | 162 [b] | 57 [c] | 103 [c] | 135 [b] | 157 [b] | 161 [b] |
| | 160 | 96 [b] | 119 [a] | 150 [a] | 176 [a] | 179 [a] | 67 [b] | 117 [b] | 152 [a] | 172 [a] | 176 [ab] |
| | 240 | 99 [a] | 121 [a] | 152 [a] | 179 [a] | 183 [a] | 77 [a] | 125 [a] | 162 [a] | 178 [a] | 183 [a] |
| Plant density (plants m$^{-2}$) | 20 | 94 [a] | 116 [a] | 147 [a] | 171 [a] | 176 [a] | 68 [a] | 117 [a] | 151 [a] | 170 [a] | 175 [a] |
| | 30 | 93 [a] | 115 [a] | 144 [a] | 171 [a] | 174 [a] | 67 [a] | 115 [a] | 150 [a] | 169 [a] | 174 [a] |
| | 40 | 93 [a] | 115 [a] | 145 [a] | 171 [a] | 174 [a] | 65 [a] | 113 [a] | 148 [a] | 168 [a] | 172 [a] |

Values followed by different small letters (a, b and c) indicate significant differences at $p < 0.05$.

The LAI under −15 kPa SMP was significantly ($p < 0.05$) higher than that under −55 kPa SMP for the two growing seasons (Table 5). No significant LAI difference occurred between −15 and −25 kPa SMP treatments for either growing season except for 76 and 89 DAS in 2018. The LAI tended to increase with the nitrogen application, and the nitrogen rate of 240 kg ha$^{-1}$ had significantly ($p < 0.05$) greater LAI than 80 and 160 kg ha$^{-1}$ except for the non-significant ($p > 0.05$) difference between 240 or 160 kg ha$^{-1}$ at 51 DAS in 2018 (Table 5). The LAI at a plant density of 40 plants m$^{-2}$ was greater ($p < 0.05$) than the LAI for 20 and 30 plants m$^{-2}$ for the entire 2018 and 2019 growing seasons (Table 5).

*3.4. The Thousand Kernel Weight and Seed Protein Content*

The thousand kernel weight increased with rising SMP in both years (Table 6). In 2018, the thousand kernel weight under −15 kPa SMP (2.28 g) was significantly ($p < 0.05$) higher than those under −25 and −55 kPa SMP treatments (2.18 and 2.12 g, respectively). Similarly, in 2019, quinoa grown with −15 kPa SMP obtained the thousand kernel weight of 2.21 g, which was significantly ($p < 0.05$) greater than those with −25 (2.11 g) and −55 (2.03 g) kPa SMP treatments. As for nitrogen application treatment, the thousand kernel weight increased significantly ($p < 0.05$) with an increase in nitrogen applications

in both years (Table 6). The thousand kernel weights reached 2.28 g in 2018 and 2.23 g in 2019 under nitrogen rate of 240 kg ha$^{-1}$, which were significantly ($p < 0.05$) greater than nitrogen rates of 80 and 160 kg ha$^{-1}$ in 2018 (2.12 and 2.18 g, respectively) and 2019 (2.02 and 2.10 g, respectively). The thousand kernel weight was not significantly affected by plant density in either year (Table 6).

**Table 5.** Effects of soil matric potential, nitrogen application rate, and plant density on quinoa leaf area index (LAI) at 51, 61, 76, and 89 days after sowing (DAS) in 2018 and 52, 61, 68, 74, and 84 DAS in 2019.

| Treatment | | 2018 | | | | 2019 | | | | |
|---|---|---|---|---|---|---|---|---|---|---|
| | | 51 | 61 | 76 | 89 | 52 | 61 | 68 | 74 | 84 |
| Soil matric potential (−kPa) | 15 | 2.5 [a] | 3.7 [a] | 6.3 [a] | 2.3 [a] | 2.6 [a] | 3.3 [a] | 5.1 [a] | 3.9 [a] | 2.6 [a] |
| | 25 | 2.6 [a] | 3.6 [a] | 5.8 [b] | 2.0 [b] | 2.4 [a] | 3.2 [a] | 4.9 [ab] | 3.8 [a] | 2.3 [a] |
| | 55 | 2.2 [b] | 2.9 [b] | 5.3 [b] | 1.6 [c] | 1.8 [b] | 2.7 [b] | 4.5 [b] | 3.3 [b] | 1.8 [b] |
| Nitrogen rate (kg ha$^{-1}$) | 80 | 2.3 [b] | 3.3 [b] | 5.2 [b] | 1.8 [c] | 2.0 [b] | 2.8 [b] | 4.3 [c] | 3.2 [c] | 1.9 [b] |
| | 160 | 2.5 [ab] | 3.4 [b] | 5.7 [b] | 1.9 [b] | 2.2 [b] | 3.1 [ab] | 4.9 [b] | 3.7 [b] | 2.2 [b] |
| | 240 | 2.6 [a] | 3.6 [a] | 6.5 [a] | 2.2 [a] | 2.6 [a] | 3.5 [a] | 5.4 [a] | 4.2 [a] | 2.7 [a] |
| Plant density (plants m$^{-2}$) | 20 | 2.2 [c] | 3.1 [b] | 5.4 [b] | 1.8 [c] | 1.9 [b] | 2.6 [c] | 4.3 [b] | 3.1 [c] | 1.8 [b] |
| | 30 | 2.4 [b] | 3.4 [b] | 5.8 [ab] | 2.0 [b] | 2.3 [a] | 3.1 [b] | 5.0 [a] | 3.6 [b] | 2.2 [b] |
| | 40 | 2.7 [a] | 3.8 [a] | 6.1 [a] | 2.1 [a] | 2.6 [a] | 3.7 [a] | 5.3 [a] | 4.3 [a] | 2.8 [a] |

Values followed by different small letters (a, b and c) indicate significant differences at $p < 0.05$.

**Table 6.** Effects of soil matric potential, nitrogen application rate, and plant density on the thousand kernel weight (g) and seed protein content (%) in 2018 and 2019.

| Treatment | | 2018 | | 2019 | |
|---|---|---|---|---|---|
| | | The Thousand Kernel Weight (g) | Protein Content (%) | The Thousand Kernel Weight (g) | Protein Content (%) |
| Soil matric potential (−kPa) | 15 | 2.28 [a] | 20.3 [ab] | 2.21 [a] | 20.4 [b] |
| | 25 | 2.18 [b] | 21.3 [a] | 2.11 [b] | 21.1 [a] |
| | 55 | 2.12 [b] | 20.0 [b] | 2.03 [c] | 19.6 [c] |
| Nitrogen rate (kg ha$^{-1}$) | 80 | 2.12 [b] | 19.8 [b] | 2.02 [c] | 19.1 [c] |
| | 160 | 2.18 [b] | 20.5 [ab] | 2.10 [b] | 20.8 [b] |
| | 240 | 2.28 [a] | 21.3 [a] | 2.23 [a] | 21.2 [a] |
| Plant density (plants m$^{-2}$) | 20 | 2.19 [a] | 20.7 [a] | 2.12 [a] | 20.6 [a] |
| | 30 | 2.18 [a] | 20.6 [a] | 2.11 [a] | 20.4 [a] |
| | 40 | 2.22 [a] | 20.4 [a] | 2.12 [a] | 20.2 [a] |

Values followed by different small letters (a, b and c) indicate significant differences at $p < 0.05$.

Results indicated that quinoa seed protein content first increased and then decreased with increasing SMP from −55, −25 to −15 kPa in both years (Table 6). Quinoa grown with −25 kPa SMP had the greatest seed protein content (21.3% in 2018 and 21.1% in 2019), significantly ($p < 0.05$ in 2018 and 2019) greater than that under −55 kPa SMP in 2018 (20.0%) and 2019 (19.6%), respectively. The difference in seed protein content between −15 and −25 kPa SMP treatments was significant ($p < 0.05$) in 2019 but not in 2018. Under nitrogen application treatment, seed protein contents for 80, 160, and 240 kg ha$^{-1}$ treatments were 19.8%, 20.5%, and 21.3% in 2018, and 19.1%, 20.8%, and 21.2% in 2019. Seed protein content did not differ significantly among plant density treatments in either year (Table 6).

*3.5. Seed Yield Per Plant*

Quinoa grown with the SMP of −55 kPa received the seed yield of 24.8−25.2 g plant$^{-1}$, which was significantly ($p < 0.05$) lower than that under −15 kPa SMP (36.2−38.1 g plant$^{-1}$) for two years (Table 3). There was no significant ($p > 0.05$) difference in seed yield between −15 and −25 kPa SMP treatments in either year. The significant difference in seed yield was found between 80 and 160 kg ha$^{-1}$ in 2019 ($p < 0.05$) but not in 2018 ($p > 0.05$), and the seed yield between 160 and 240 kg ha$^{-1}$ did not differ

significantly ($p > 0.05$) in either year (Table 3). The seed yield significantly ($p < 0.05$) decreased with increasing plant density from 20 to 40 plants m$^{-2}$, and the seed yield of 20 plants m$^{-2}$ (13.5 g plant$^{-1}$) was 6.3 and 13.5 g plant$^{-1}$ higher than that under 30 and 40 plants m$^{-2}$, respectively (Table 3).

*3.6. Dry Matter and Nitrogen Content of Leaf, Stem, and Seed, and Nitrogen Uptake*

The soil matric potential affected the dry matter of leaf, stem, and seed significantly ($p < 0.05$) in both years (Table 7). The −15 kPa SMP treatment obtained the significantly ($p < 0.05$) greater leaf (12.3–12.4 Mg ha$^{-1}$), stem (33.1–34.1 Mg ha$^{-1}$), and seed dry matter (10.9–11.4 Mg ha$^{-1}$) than −55 kPa SMP treatment (leaf dry matter of 4.0–4.2 Mg ha$^{-1}$, stem dry matter of 13.2–13.9 Mg ha$^{-1}$, and seed dry matter of 7.4–7.7 Mg ha$^{-1}$). Except for the −25 and −55 kPa SMP treatments in 2018, there was no significant ($p > 0.05$) difference in leaf or stem protein content among SMP treatments in either year. Leaf dry matter did not differ significantly ($p > 0.05$) among nitrogen rate treatments in either year; however, stem dry matter increased with an increasing nitrogen rate, and the differences were significant ($p < 0.05$) between 80 and 240 kg ha$^{-1}$ treatments in both years (Table 7). Nitrogen rate had no significant ($p > 0.05$) effect on stem protein content in either year, and the effects of nitrogen rate were significant on leaf protein content in 2018 ($p < 0.05$) but not in 2019 ($p > 0.05$). Dry matter of leaf and stem remained unaffected by plant density in either year ($p > 0.05$), whereas the difference in seed dry matter between 20 and 40 was significant ($p < 0.05$) in both years. As the plant density increased, there was a decrease in leaf nitrogen content, and the difference was significant between 20 and 40 plants m$^{-2}$ treatments ($p < 0.05$). The differences in stem protein content between 20 and 40 plants m$^{-2}$ treatments were significant in 2018 ($p < 0.05$) but not in 2019 ($p > 0.05$) (Table 7).

**Table 7.** Effects of soil matric potential, nitrogen application rate, and plant density on the dry matter (Mg ha$^{-1}$) and nitrogen content (%) of leaf, stem, and seed, and nitrogen uptake (kg ha$^{-1}$) in 2018 and 2019.

| Treatment | | Dry Matter (Mg ha$^{-1}$) | | | Nitrogen Content (%) | | | N Uptake (kg ha$^{-1}$) |
|---|---|---|---|---|---|---|---|---|
| | | Leaf | Stem | Seed | Leaf | Stem | Seed | |
| | | 2018 | | | | | | |
| Soil matric potential (−kPa) | 15 | 12.4 [a] | 34.1 [a] | 11.4 [a] | 1.8 [a] | 0.8 [ab] | 3.2 [ab] | 857 [a] |
| | 25 | 8.0 [ab] | 24.9 [b] | 9.4 [ab] | 1.9 [a] | 0.7 [b] | 3.4 [a] | 647 [b] |
| | 55 | 4.0 [b] | 13.9 [c] | 7.7 [b] | 1.8 [a] | 0.9 [a] | 3.2 [b] | 438 [c] |
| Nitrogen rate (kg ha$^{-1}$) | 80 | 6.1 [a] | 18.5 [c] | 8.9 [a] | 1.6 [b] | 0.8 [a] | 3.2 [b] | 536 [b] |
| | 160 | 7.9 [a] | 24.8 [b] | 9.8 [a] | 1.9 [ab] | 0.8 [a] | 3.3 [ab] | 657 [ab] |
| | 240 | 10.4 [a] | 29.6 [a] | 9.7 [a] | 2.0 [a] | 0.8 [a] | 3.4 [a] | 748 [a] |
| Plant density (plants m$^{-2}$) | 20 | 6.9 [a] | 22.5 [a] | 7.9 [b] | 2.0 [a] | 0.9 [a] | 3.2 [a] | 602 [a] |
| | 30 | 7.9 [a] | 23.6 [a] | 10.0 [ab] | 2.0 [a] | 0.8 [b] | 3.3 [a] | 666 [a] |
| | 40 | 9.6 [a] | 23.9 [a] | 10.5 [a] | 1.5 [b] | 0.7 [b] | 3.3 [a] | 674 [a] |
| | | 2019 | | | | | | |
| Soil matric potential (−kPa) | 15 | 12.3 [a] | 33.1 [a] | 10.9 [a] | 1.7 [a] | 0.6 [a] | 3.3 [b] | 817 [a] |
| | 25 | 7.7 [ab] | 22.9 [ab] | 9.8 [a] | 2.0 [a] | 0.8 [a] | 3.4 [a] | 628 [b] |
| | 55 | 4.2 [b] | 13.2 [b] | 7.4 [b] | 1.8 [a] | 0.8 [a] | 3.1 [c] | 514 [c] |
| Nitrogen rate (kg ha$^{-1}$) | 80 | 6.3 [a] | 18.1 [b] | 8.5 [b] | 1.7 [a] | 0.8 [a] | 3.1 [b] | 515 [c] |
| | 160 | 7.2 [a] | 22.4 [ab] | 9.8 [a] | 2.0 [a] | 0.7 [a] | 3.3 [b] | 620 [b] |
| | 240 | 10.6 [a] | 28.8 [a] | 9.7 [a] | 1.8 [a] | 0.7 [a] | 3.4 [a] | 725 [a] |
| Plant density (plants m$^{-2}$) | 20 | 6.6 [a] | 21.5 [a] | 7.9 [b] | 2.0 [a] | 0.7 [a] | 3.3 [a] | 552 [b] |
| | 30 | 7.9 [a] | 21.8 [a] | 9.9 [ab] | 2.0 [ab] | 0.8 [a] | 3.3 [a] | 643 [ab] |
| | 40 | 10.5 [a] | 26.0 [a] | 10.3 [a] | 1.5 [b] | 0.8 [a] | 3.2 [a] | 666 [a] |

Values followed by different small letters (a, b and c) indicate significant differences at $p < 0.05$.

Quinoa grown under −15 kPa SMP obtained the highest nitrogen uptake (857 kg ha$^{-1}$ in 2018 and 817 kg ha$^{-1}$ in 2019), which was significantly ($p < 0.05$) higher than at −25 kPa (647 kg ha$^{-1}$ in 2018 and 628 kg ha$^{-1}$ in 2019) and −55 kPa (441 kg ha$^{-1}$ in 2018 and 415 kg ha$^{-1}$ in 2019) (Table 7). Nitrogen

uptake increased significantly ($p < 0.05$) with a higher nitrogen application rate (Table 7). The nitrogen uptake under a nitrogen rate of 240 kg ha$^{-1}$ (738 and 725 kg ha$^{-1}$ in 2018 and 2019, respectively) was significantly ($p < 0.05$) greater than those under 80 kg ha$^{-1}$ (738 kg ha$^{-1}$ in 2018 and 725 kg ha$^{-1}$ in 2019). The nitrogen rate of 240 kg ha$^{-1}$ had numerically greater nitrogen uptake than the 160 kg ha$^{-1}$ nitrogen rate in 2018 and significantly ($p < 0.05$) greater nitrogen uptake than the 160 kg ha$^{-1}$ rate in 2019. Higher plant densities resulted in greater nitrogen uptake (Table 7). Nitrogen uptake under the plant density of 40 plants m$^{-2}$ was significantly ($p < 0.05$) greater than 20 plants m$^{-2}$. There was no significant ($p > 0.05$) difference in nitrogen uptake between plant densities at 30 and 40 plants m$^{-2}$ (Table 7).

## 4. Discussion

The −15 kPa SMP irrigation onset criteria treatment maintains relatively constant and high soil moisture (above 77% field capacity) in the root zone, whereas the −55 kPa SMP irrigation onset criteria treatment causes larger fluctuations of the soil water content in the root zone between two irrigation events and greater water stress when SMP reaches its threshold value, possibly causing adverse soil water conditions (43% field capacity) that limit the uptake of water and nitrogen and limit crop growth [17,28,44]. Not surprisingly, the significantly ($p < 0.05$) lowest plant height, LAI, $ET_C$ (311–347 mm), dry matter, seed yield per plant (24.8–25.6 g plant$^{-1}$), seed protein content (19.6%–20.0%), and thousand kernel weight (2.03–2.12 g) were all observed under −55 kPa SMP, consistent with some early reports on quinoa [19,45,46], suggesting that a soil matric potential of −55 kPa results in notable water stress for quinoa growth and development. Similar to the reports for many other crops (corn, potato, and chili pepper) [29,44], quinoa seed yield did not differ significantly ($p > 0.05$) between −15 and −25 kPa SMP treatments in both years, suggesting that crop yields would not be greatly affected under an SMP around −25 kPa (60% field capacity). Quinoa grown with an irrigation onset at −15 kPa SMP had significantly ($p < 0.05$) greater thousand seed weight (2.21–2.28 g) than at −25 kPa SMP (2.11–2.18 g), which might be caused by the high sensitivity of quinoa seed weight to water stress, similar with the results of Ince Kaya and Yazar (2016) [47]. Additionally, seed protein content significantly ($p < 0.05$) decreased when SMP increased from −25 (21.1%–21.3%) to −15 (20.3%–20.4%) kPa. These results are in agreement with those reported by Oktem [48], but different from those obtained by Walters et al. [46]. However, our study indicated that slight water stress of −25 kPa SMP improved quinoa seed protein content, consistent with Oktem [48]. According to the above results, an irrigation onset criteria of SMP ranging from −15 to −25 kPa is recommended to schedule drip irrigation for enhancing quinoa yield and quality.

Higher nitrogen fertilizer applications promoted plant growth and canopy development, resulting in greater plant height and LAI, in accordance with the results of Elbehri et al. [49], Jacobsen et al. [50], and Alandia et al. [51]. With a nitrogen application rate increasing from 80 to 160 kg ha$^{-1}$, seed yield increased by 10%–15% but showed no further increase with nitrogen application up to 240 kg ha$^{-1}$. Increasing the nitrogen application rate to 240 kg ha$^{-1}$ caused significant ($p < 0.05$) increases in the thousand seed weight and seed protein content, up to 2.26 g and 21.3%, indicating that the quinoa seed protein could be strongly influenced by nitrogen supply and might be further enhanced under higher nitrogen rate [9].

Seed yield per plant decreased with increasing plant density due to the fierce competition for light and nutrients among individuals [34,52]. The plant density of 40 and 30 plants m$^{-2}$ was 2.0 and 1.5 times the plant density of 20 plants m$^{-2}$, respectively; however, the seed yield of 20 plants m$^{-2}$ (39.5 g plant$^{-1}$) was only 1.5 and 1.2 times the seed yield under 40 (26.0 g plant$^{-1}$) and 30 (33.2 g plant$^{-1}$) plants m$^{-2}$, indicating that the overall seed yield could be greater with plant density beyond 20 plants m$^{-2}$ in quinoa production, comparable to the results of seed dry matter in this study. In addition, the plant density of 40 plants m$^{-2}$ was 1.3 times the 30 plants m$^{-2}$, and the seed yield under 30 plants m$^{-2}$ was also nearly 1.3 times that under 40 plants m$^{-2}$, respectively, suggesting that the overall yield would

be similar between 30 and 40 plants m$^{-2}$. Plant density did not affect seed quality (seed weight and protein content) significantly ($p > 0.05$), consistent with previous results [34,35].

In our experiments, the 240 kg ha$^{-1}$ nitrogen fertilizer application treatment had the maximum $ET_C$ (420–455 mm), followed by 160 kg ha$^{-1}$ (413–430 mm) and 80 kg ha$^{-1}$ (398–429 mm), indicating that more nitrogen fertilizer application leads to greater water consumption, as previously reported [51]. Higher SMP positively affects the capacity of plants for nitrogen uptake [24], and the significantly ($p < 0.05$) greater nitrogen uptake under −15 kPa (817–857 kg ha$^{-1}$) than −55 kPa (415–441 kg ha$^{-1}$) was found in our experiments. In addition, quinoa under higher plant density had more $ET_C$ and nitrogen uptake, peaking at 440–479 mm and 666–697 kg ha$^{-1}$ at 40 plants m$^{-2}$, consistent with previous results [53–55].

Quinoa seed protein content that ranged from 19.1% to 21.3% in our experiments was within the range of quinoa protein content of 7.47%–22.08% reported by Cardozo and Tapia (1979) [56] and 7%–24% reported by Koziol (1992), Wright et al. (2002), Repo-Carrasco et al. (2003), and Bhargava (2007) [57–60]. Furthermore, the protein content of stem and leaf was similar to the findings of in an earlier report [5]. However, the dry matter in our experiments was greater than some former results [23,35]. The greater dry matter in our experiments should result from the proper management of irrigation, nitrogen rate, and plant density (especially under −15 kPa SMP, nitrogen rate of 240 kg ha$^{-1}$, and plant density of 40 plants m$^{-2}$). Therefore, our results indicated that the calculated amount of nitrogen uptake by quinoa (410–860 kg ha$^{-1}$) was far greater than the applied nitrogen (80–240 kg ha$^{-1}$), and significantly increased with the combined effects of sufficient water, nitrogen supply, and high plant density.

The available soil nitrogen before sowing in our experiment was 774 kg ha$^{-1}$ in 2018 and 927 kg ha$^{-1}$ in 2019 (Equation (4)). That is to say, the plant consumed most of the available nitrogen in the soil to meet its requirements, especially under the high nitrogen uptake treatments, suggesting that quinoa should be considered as a rotational crop to maintain the balance of soil nitrogen [14,15,61].

## 5. Conclusions

The SMP irrigation onset threshold of drip irrigation within the range from −15 to −25 kPa increased the yield per plant by 47% to 48% than the drier treatment at an SMP of −55 kPa. Furthermore, quinoa grown under an SMP of −15 to −25 kPa also obtained greater thousand seed weight (2.20 g on average) and protein content (20.8% on average) than −55 kPa SMP treatment (2.08 g and 19.8% on average, respectively). Therefore, quinoa should be irrigated in a timely manner (irrigation was applied when the soil matric potential immediately reached −15–−25 kPa to obtain great seed yield and quality).

The plant height, LAI, and dry matter all increased with an increasing nitrogen rate from 80 kg ha$^{-1}$ to 240 kg ha$^{-1}$, but the seed yield per plant did not further increase when the nitrogen rate was beyond 160 kg ha$^{-1}$. The 240 kg ha$^{-1}$ treatment achieved the significantly ($p < 0.05$) greatest thousand seed weight (2.26 g on average) and protein content (21.3% on average), implying that seed quality might be further increased with a higher nitrogen rate.

Increasing plant density increased LAI, dry matter, and uptake of water and nitrogen. The seed yield per plant under 20 plants m$^{-2}$ was 24% and 52% greater than those under 30 and 40 plants m$^{-2}$, respectively. The overall seed yield might increase when plant density was beyond 20 plants m$^{-2}$, and it might be similar between 30 and 40 plants m$^{-2}$. Considering the greater uptake of water and nitrogen under higher plant density, the plant density of 30 plants m$^{-2}$ was recommended for quinoa cultivation.

It is worth noting that the plant consumed most of the available nitrogen in the soil (410–860 kg ha$^{-1}$ in our experiments), especially with sufficient water, nitrogen supply, and high plant density, suggesting that quinoa has an astonishingly high nitrogen requirement and should be cultivated in a crop rotation system to maintain the balance of soil nitrogen.

**Author Contributions:** Conceptualization, N.W. and F.W.; Methodology, N.W., F.W., C.M., and L.Q.; Investigation, N.W.; Resources, F.W.; Writing—original draft preparation, N.W.; Writing—review and editing, F.W. and C.C.S.; Supervision, F.W.; Project administration, F.W.; Funding acquisition, F.W. All authors have read and agree to the published version of the manuscript.

**Funding:** This research was funded by the Ministry of Water Resources of China, grant number 201501017, the Foundation for Innovative Research Groups of the National Natural Science Foundation of China, grant number 51621061, and the Major Program of the National Natural Science Foundation of China, grant number 51439006.

**Acknowledgments:** We wish to thank the staff of the Shiyang River Basin Experimental Station for Agricultural and Ecological Water Saving of China Agricultural University.

**Conflicts of Interest:** All authors declare no conflict of interest.

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
