# Peer review of "Effects of Management Practices on Quinoa Growth, Seed Yield, and Quality"

_agronomy, doi:10.3390/agronomy10030445_

Round 1

Reviewer 1 Report

The manuscript has been improved a lot, and all suggestions and comments have been considered.

I thanks the authors for providing references supporting their position and statements, I found them very useful.  

Reviewer 2 Report

Improvements have been done, nevertheless, major issues are still observed in the manuscript.

Abstract:
Line 13: what do you mean by well-managed practices? Write good management practices, instead.
Line 19: Despite the previous revision, authors insist to include this yield estimation that is not correct. Considering the plot size you had, you should have taken the yield of 1 m2 and from there the estimation is possible, but from only 15 plants, this estimation cannot be correct. This should be changed and only the yields per plant should be included. Yields become highly questionable when authors have used less than half of the recommended density for this crop.
Introduction:
Line 31: what is modest management? Be concise.
Lines 32 – 33: The recommended management practices should be clarified, without correct ranges, the introduction is vague.
Methods:
Line 102: What is middle maturity in China? this should be specified in days of growing season
Line 103: The origin of genotype Longli #1 and reference used should be revised, the information is not correct. Knowing how this material was bred provides explanations on the response to your treatments. Whether this material was just adapted or resulted from crosses of other parental material helps to understand the results.
Lines 104-106: unclear sentence.
Lines 146-147: natural air or oven drying method?
Lines 147 – 149: this estimation done with a sampling of 15 plants cannot be correct.
Section 2.6 needs to be re-written. Is not clear. Factors are mixed together with variables, ANOVA description is repetitive and does not say much. How was the effect of blocks analyzed?
Line 157: why 6.25? what is that? Why do you choose to use that?
Results:
Fig 2 very low quality

Section 3.5: All section needs to be re-written excluding all the estimations of Mg/ha which are not correct.

Until the results are not reported and discussed in terms of grams/plant, further revision cannot take place.

Author Response

This manuscript is a resubmission of an earlier submission. The following is a list of the peer review reports and author responses from that submission.

Round 1

Reviewer 1 Report

In general the manuscript is well written and discussed but I have many concerns about references cited and numbers given in the results. In particular, the seed yields of this study have never been reported elsewhere. Authors mention yields up to 12.8 Mg ha-1 while literature (also the cited one) extensively reports values below 3.5 Mg ha-1 (under many conditions, locations and experiments).

In the same way, seed protein content seems really high, around 20% when literature normally reports values around 11-15%.

N uptake is also not in line with existing studies as it is here reported up to 857 kg ha-1 while it is normally about a half in the best conditions.

Finally, checking the reported literature, there are also some problems:

Line 29: reviewer was not able to verify the information provided in those references. Reference 6 is not available and in a different language, Reference 8 does not mention 4600-6000 kg ha-1.

Line 30: reference 8. Schulte et al., (2005) don’t use 2200 and 3500 kg ha-1 of nitrogen. However, the N rates examined Schulte et al., is 0, 80 and 120 kg N ha-1.

Line 30: reference 10. Risi and Galwey (1991) examine plant density using a maximum of 15 kg ha-1 of seeds not 3000-6300 kg ha-1as mentioned in this research.

Line 34: (correct) reference 15. Geerts et al., (2008) see a positive correlation between seed yield weight and greater irrigation. In their research, T1, T4 and T7 (under full irrigation) show the highest kernel weight.

Line 49: in Egypt (reference 30) the highest yields are obtained with 310 kg N ha-1 rather than with 430 kg N ha-1

Materials and Methods:

Lines 123-124: LAI was measured in different dates (55, 61, 76 and 89 DAS for 2018 and 52, 61, 68, 74 and 84 DAS for 2019), making it difficult the comparison between years. In particular (as seen in results section), when such large differences are observed in few days in terms of LAI and plant height.

Lines 131: even if known, please explain to readers why is the WUE equation divided by 10

Results:

Line 171: Table 3: it’s enough by indicating significant difference at P < 0.05. Too many letters make it hard to read. 

Line 171: Table 3: does not show significant differences on yield between difference nitrogen rates in 2018. Nonetheless, some differences between 240 and 160 kg N ha-1 when compared to 80 kg N ha-1

Line 180-181: it’s stated that the “nitrogen rate of 240 kg ha-1 resulted in the tallest (P < 0.05) quinoa plants, followed by 160 kg ha-1and 80 kg ha-1”. However, for 2018 there are no differences between 160 and 240 kg N ha-1. The only treatment that has significant differences with the rest is 80 kg N ha-1

Line 184: Table 4: plant height units are not specified.

Line 184: For 2018, is it physiologically possible to have a plant height increase of up to 25 cm in just three days (49 DAS and 52 DAS)?

Line 184: It’s enough by indicating significant difference at P < 0.05

Lines 242-253: a nitrogen-uptake of 857 kg ha-1 is extremely high. These values are closer to N-balance (changes in soil N content before sowing and after harvesting, including inputs from fertilization and atmospheric deposition and outputs from volatilization, plant uptake and others) rather than N-uptake. In particular, because your nitrogen availability in the soil before sowing is of approximately150 kg N ha-1. Please verify.